# Effect of a Total Extract and Saponins from *Astragalus glycyphyllos* L. on Human Coronavirus Replication In Vitro

**DOI:** 10.3390/ijms242216525

**Published:** 2023-11-20

**Authors:** Anton Hinkov, Venelin Tsvetkov, Aleksandar Shkondrov, Ilina Krasteva, Stoyan Shishkov, Kalina Shishkova

**Affiliations:** 1Laboratory of Virology, Faculty of Biology, University of Sofia “St. Kl. Ohridski”, 1164 Sofia, Bulgaria; ven_tsvetkov@abv.bg (V.T.); sshishkov@biofac.uni-sofia.bg (S.S.); 2Department of Pharmacognosy, Faculty of Pharmacy, Medical University of Sofia, 2 Dunav St., 1000 Sofia, Bulgaria; shkondrov@pharmfac.mu-sofia.bg (A.S.); ikrasteva@pharmfac.mu-sofia.bg (I.K.)

**Keywords:** human coronavirus 229E, *Astragalus glycyphyllos* L., antiviral activity, saponin

## Abstract

Members of the family Coronaviridae cause diseases in mammals, birds, and wildlife (bats), some of which may be transmissible to humans or specific to humans. In the human population, they can cause a wide range of diseases, mainly affecting the respiratory and digestive systems. In the scientific databases, there are huge numbers of research articles about the antiviral, antifungal, antibacterial, antiviral, and anthelmintic activities of medicinal herbs and crops with different ethnobotanical backgrounds. The subject of our research is the antiviral effect of isolated saponins, a purified saponin mixture, and a methanol extract of *Astragalus glycyphyllos* L. In the studies conducted for the cytotoxic effect of the substances, CC_50_ (cytotoxic concentration 50) and MTC (maximum tolerable concentration) were determined by the colorimetric method (MTT assay). The virus was cultured in the MDBK cell line. As a result of the experiments carried out on the influence of substances on viral replication (using MTT-based colorimetric assay for detection of human Coronavirus replication inhibition), it was found that the extract and the purified saponin mixture inhibited 100% viral replication. The calculated selective indices are about 13 and 18, respectively. The obtained results make them promising for a preparation with anti-Coronavirus action.

## 1. Introduction

Respiratory viruses are the main cause of upper and lower respiratory tract infections. They cause high morbidity worldwide. Infectious Coronavirus disease 2019 (COVID-19) is a new respiratory disease caused by Coronavirus 2 (SARS-CoV-2). The Coronaviridae family includes enveloped viruses with a spherical shape. The size of the virion varies between 80 and 220 nm in diameter. The genome is single-stranded (+) RNA. The Coronaviridae family includes three subfamilies with a total of six genera. Members of the family cause diseases in mammals, birds, and wildlife (bats), some of which may be transmissible to humans or specific to humans. In the human population, they can cause a wide range of diseases, mainly affecting the respiratory and digestive systems. What the representatives of the family have in common is that they are highly transmissible. Seven members of the family have been found to cause disease in humans. Human Coronavirus (HCoV)-NL63, -229E, -HKU1, and -OC43 usually cause mild-to-moderate respiratory diseases with a seasonal pattern. SARS-CoV, MERS-CoV, and SARS-CoV-2 can be responsible for severe lung damage and respiratory distress syndrome that can cause death in humans. At the end of 2019, Wuhan, a major business center in China, experienced the emergence of a new Coronavirus, i.e., SARS-CoV-2 (COVID-19), which is described as a subtype of the β subgroup of Coronaviruses [1]. SARS-CoV-2 has dramatically spread to more than 200 countries and territories worldwide, infecting an estimated two million people worldwide. This virus not only attacks various organs of the body, including the respiratory system, the gastrointestinal system, and the liver, but it also invades the central nervous system and causes disease in both humans and animals, such as bats, cattle, birds, and mice [2].

There is evidence that the emerging virus outbreak is due to zoonotic transmission, as it belongs to the same family of viruses as civet-related SARS-CoV and camel-related MERS-CoV. The outbreak of the COVID-19 pandemic reminds us that mutations in Coronaviruses can sometimes allow host switching. It was described by Shi et al. that SARS-CoV-2 shares 96.2% similarity with the SARS-CoV Coronavirus that originated from Rhinolophus affini (bats) [1]; this clearly indicates that the emerging virus may also originate from bats.

At the infection stage, the virus attaches itself to the cell with a spike glycoprotein (protein S), which is located in the outer envelope. This protein consists of two subunits responsible for cell attachment and subsequent fusion [1,3,4]. The target of the binding is the cellular membrane protein, angiotensin-converting enzyme 2 (ACE2). The ability to bind the virus to the ACE2 membrane receptor is much stronger for SARS-CoV-2 compared to SARS-CoV, which has the same binding site [4,5,6]. Protein S is hydrolyzed by endosomal proteases (cathepsin or serine protease 2 (TMPRSS2)), which results in membrane fusion [7]. Both the receptor itself and the proteases cutting spike protein are potential therapeutic targets. 

In the fight against viral infections, various medications are used, the most commonly used being nucleoside analogs. It is important to have potent and safe drugs at hand that can be used for the treatment or prophylaxis of such infections. Among the large number of highly active medicinal preparations created, there are also those that have numerous unwanted side effects on the body. Therefore, phytoproducts and biologically active substances contained in plants are of increasing interest. Plant-natural products or extracts have been used in folk medicine for hundreds of years to treat viral diseases [8]. They are much better tolerated by the human body and are often less toxic. Medicinal plants have by no means lost their importance despite the increase in the relative share of synthetic preparations. Total plant extracts mimic combination therapy with several synthetic agents (a major approach to overcoming the emergence of drug resistance) because they contain more than one biologically active substance.

In the scientific databases, there are huge numbers of research articles about the antiviral, antifungal, antibacterial, antiviral, and anthelmintic activities of medicinal herbs and crops with different ethnobotanical backgrounds [9,10,11,12]. Connections between the health benefits of medicinal plants and the presence of specific secondary metabolites, which can be responsible for some effects, are also described. It was suggested that terpenoids, alkaloids, stilbenes, and flavonoids are the main biologically active compounds for the development of antiviral plant-based drugs [13].

Natural products have been in constant use since ancient times and have proven over time to be effective. Crude extract or pure compounds isolated from medicinal plants and/or herbs, such as *Artemisia annua* L., *Agastache rugosa* (Fisch. and C.A.Mey.) Kuntze, *Astragalus membranaceus* L., *Cassia alata* L., *Ecklonia cava* Kjellman, *Gymnema sylvestre* R. Br., *Glycyrrhizae uralensis* Fisch. ex DC., *Houttuynia cordata* Thunb., *Lindera aggregate* (Sims) Kosterm., *Lycoris radiata* (L’Hér.) Herb., *Mollugo cerviana* (L.) Ser., *Polygonum multiflorum* Thunb., *Pyrrosia lingua* (Thunb.) Farw., *Saposhnikoviae divaricate* (Turcz.) Schischk., and *Tinospora cordifolia* (Thunb.) Miers, have shown promising inhibitory effects against Coronavirus. Several molecules, including acacetin, amentoflavone, allicin, blancoxanthone, curcumin, daidzein, diosmin, epigallocatechin-gallate, emodin, hesperidin, herbacetin, hirsutenone, iguesterin, jubanine G, kaempferol, lycorine, pectolinarin, phloroeckol, silvestrol, tanshinone I, taxifolin, rhoifolin, xanthoangelol E, and zingerol, isolated from plants could also be potential drug candidates against SARS-CoV-2 [14]

Herbal extracts from the traditional Chinese medicine plants *Cibotium barometz* (L.) J.Sm., *Gentiana scabra* Bunge, *Dioscorea batatas* Decne., *Cassia tora* (L.) Roxb., and *Taxillus chinensis* (DC.) Danser were found to inhibit SARS-CoV replication [15]. Extracts of *Sophorae flos*, *Acanthopanacis cortex*, *Sanguisorbae radix*, and *Torilis fructus* possess antiviral effects that are attributed to the inhibition of RNA-dependent RNA polymerase or protease activity required for RNA replication [16]. Herbal extracts of *Cimicifuga rhizomes*, *Meliae cortex, Coptidis rhizoma*, and *Phellodendron cortex* also affected Coronavirus replication [17].

*Astragalus* extracts and their isolated components exhibited promising in vitro and in vivo biological activities, one of which is an antiviral action [8]. *Astragali radix* extracts show protective effects against Japanese Encephalitis Virus (JEV) infection in mice [18]. *Astragalus* polysaccharides (APS) from *Astragalus mongholicus* Bunge have shown in different studies that they can inhibit the replication of several animal viruses, including H9N2 avian influenza virus, foot and mouth disease virus, Newcastle disease virus, and infectious bursal disease virus [19,20].

In view of the above and due to the fact that our laboratory has a tradition of studying the antiviral effect of plant extracts, we aimed to study an extract of *Astragalus glycyphyllos* L. to establish its effect on the replication of human Coronavirus 229E (HCoV-229E). This virus causes respiratory disease and belongs to the α subgroup of Coronaviruses. The aim of this study is the possible antiviral activity of a standardized extract of the aerial parts of the species. To date, despite the wide use of *A. glycyphyllos* in the traditional medicine of many countries, reports on its antiviral effects are very limited [21].

## 2. Results

### 2.1. Phytochemistry of DEAG, PSM, and Saponins S1, S3, S9

A dry extract was obtained (62 g) from the overground parts of *A. glycyphyllos* L., comprising 21% of the plant material. The UHPLC–HRESIMS qualitative analysis revealed that DEAG contained camelliaside A, mauritianin and rutin, and triterpenoid saponins (S1, S3, and S9) [22]. Representative chromatograms are given in Appendix A. In DEAG, the main saponin (S1) was further used for the quantitative analytical marker. The quantitative analysis established that DEAG contained 3.12% total saponins, expressed as S1 [23]. The assay of PSM showed that it had 50.52% total saponins as the same marker [23]. Chromatograms of PSM are given in Appendix A. The structures of the three main saponins are presented in Figure 1. The three saponins were isolated from the original plant source, as reported before [24,25]. They had the following purity (UHPLC–HRESIMS): S1—98%, S3—98%, and S9—97%. Chromatograms of the compounds are presented in Appendix A.

### 2.2. Cultivation of Human Coronavirus 229E in Cell Culture

The first manifestations of a cytopathic effect were observed on the second day after inoculation of the monolayer. Fields of clustered cells with altered morphology (reduced size and jagged outline) were observed. Four days after the inoculation of the cell culture, all of its cells were infected, and the cytopathic effect unfolded and was observed throughout the cell monolayer. The cytopathic effect was expressed in the shrinking of cells (reduction in their diameter), loss of adhesive ability, and detachment of cells from the bottom of the culture vessel. Some of the cells were lysed. The resulting virus suspension was quantitatively characterized by the final limiting dilution method, and the infectious titer (which is 6.57 TCID_50_/_mL_ (tissue culture infectious dose)) was calculated by the Reed–Muench method [26]. The results of virus cultivation and the occurrence of cytopathic effect are presented in Figure 2.

### 2.3. Cytotoxicity Assay

After the isolation of the virus, the cell viability of the cell line used was determined under the influence of DEAG, the purified saponin mixture, and the individual saponins by MTT test. Toxicity was determined at the 72nd hour. The maximum tolerated concentration was determined both by the MTT test and microscopically. In the experimental setup used, the substances were administered in concentration ranges from 3 mg/mL to 0.000728 mg/mL. 

When the results were reported (Table 1), the MTC (maximum tolerable concentration) for DEAG was found to be 1 mg/mL, and the CC_50_ (cytotoxic concentration 50) value was 1.272 mg/mL. The saponin mixture and isolated saponins exhibited higher toxicity. The reported MTC value for PSM and C3 is 0.025 mg/mL. For saponins S1 and S9, the MTC value is 0.00625 mg/mL. The determination of MTC is carried out both microscopically at the 48th and 72nd hours, as well as by the MTT test. In terms of CC_50_, PSM exhibits the highest toxicity.

### 2.4. Antiviral Assay

When conducting the experiments on the influence of substances on viral replication, the same was applied in MTC and several lower than it. When the total extract was applied in MTC in both experimental setups (see Materials and Methods), the percentage of protection of the infected cells was 100. The same percentage of protection was maintained up to a concentration of the applied extract of 0.2 mg/mL. When the extract was added simultaneously with the infection of the cells, the determined effective concentration_50_ (EC_50_) was 0.091 mg/mL. The EC_50_ is the concentration of the extract that inhibits viral replication by 50% and is determined graphically from the dose–response curve. The calculated selectivity index (SI) using this experimental setup (the ratio of CC_50_ to EC_50_) was 13.97. In the second experimental setup, in which the extract in MTC was added 1.5 h after virus adsorption, the MTT assay results again showed 100 percent inhibition of viral replication. The effect is maintained when the extract is administered in several successive concentrations lower than MTC. In MTC, as well as in most of the used concentrations, DEAG achieved 100%. The determined effective concentration was 0.114 mg/mL. The calculated SI is 11.15. The results are depicted graphically in Figure 3a. 

When the PSM was administered in MTC, the same percentage of protection of the infected cells was found. The calculated selectivity index was comparable using both trial settings and was higher compared to those calculated for DEAG in the same experimental settings. The results are presented graphically in Figure 3b. 

The individual saponins show a weak antiviral effect, and the calculation of a selective index is not possible. 

These results indicate that DEAG and PSM mainly affect the intracellular stages of viral replication (antiviral activity is not dependent on the time of administration). The results of the conducted experiments are presented in Table 1.

From the obtained results, it could be concluded that the total extracts of medicinal plants could show a better effect due to their rich component composition and possibly the synergism between the individual components. In support of the above is the fact that the isolated saponins do not show an effect, unlike the PSM and DEAG. 

## 3. Discussion

Plants are a rich source of substances with biological activity. The lack of literature data on the antiviral activity of *A. glycyphyllos*, its use in traditional medicine, and the available information on its phytochemical constituents prompted us to investigate the effect of DEAG, a purified saponin mixture, and isolated saponins. In previous studies by the author’s team, we found that DEAG protects those infected with SvHA1 (F strain) and SvHA2 (DD strain), with the selectivity index varying between 6 and 9. The antiviral effect of individual saponins was also investigated separately, but unfortunately, no significant findings were observed. Based on the results obtained, tablets were prepared from the standardized dry extract [21]. We focused on the saponin mixture, as saponins’ neuroprotective, antioxidant, immunomodulating, antiproliferative, antitumor, and hMAO-B-inhibiting effects have been investigated and proven in previous studies [24,25,27,28]. Several studies have demonstrated the cytoprotective effects of flavonoids and saponins isolated from the *Astragalus* species [29]. In the present study, we used DEAG, three isolated saponins, and a purified saponin mixture to establish anti-Coronavirus activity.

Usually, the VERO 6 cell line is used for the cultivation of this virus. For the cultivation of HCoV-229E, the MDBK cell line was used in our experiments in order to compare the effect of the total extract on the replication of herpes viruses and the replication of HCoV-229E using the same cell culture. It turned out that a productive infection with a distinct cytopathic effect and the possibility of isolating the virus could also take place in this cell line. We assume that the reason for this fact is the presence of an AT4 receptor in epithelial cells [30] from the kidney of Madin–Darby cattle (cell line MDBK) and, accordingly, the possibility of entering the virus by binding to it, which with the information available at this stage about Coronavirus receptors seems interesting to us. After the isolation of the virus from the cell culture, experiments were carried out on the influence of the substances on the replication of the virus.

When summarizing the obtained results, we found that the purified saponin mixture and DEAG protect 100% virus-infected cells, and the results are almost identical when applying the two experimental setups. The selective index when treating the infected cells with the total extract is between 11 and 14, and PSM treatment is between 17 and 18. The fact that there is no difference when applying the two experimental setups (simultaneously and consequently) indicates that the substances affect viral replication and not the extracellular form of the virus. The individual saponins again did not show antiviral activity against the Coronavirus we used.

## 4. Materials and Methods

### 4.1. Preparation of the Dried Extract (DEAG)

The overground parts of *A. glycyphyllos* were collected during the flowering stage in June 2022 from a locality in Vitosha Mountain, Sofia region, Bulgaria. The species was identified by Dr. D. Pavlova from the Faculty of Biology, Sofia University, where a voucher specimen was deposited (SO-107613). The air-dried pulverized (3 mm) plant material (300 g) was extracted with 50% MeOH and lyophilized, as reported in [21]. It was named DEAG and used for the experiments. All reagents and solvents are obtained from Sigma-Aldrich, Darmstadt, Germany and were of HPLC grade.

### 4.2. Preparation of the Purified Saponin Mixture (PSM)

The extract was separated by column chromatography (CC) over Diaion HP-20 (Mitsubishi Chemicals, Tokyo, Japan), eluting with a gradient of water–methanol as previously described [24]. Eight fractions were obtained, as by TLC analysis (Silica gel plates, EtOAC:HCOOH:AcOH:H_2_O = 32:3:2:6, anisaldehyde/c. H_2_SO_4_, 104 °C, 10 min) (Merck, Darmstadt, Germany), the seventh fraction was found rich in saponins [26]. It was named PSM and used for the experiments. All reagents and solvents are obtained from Sigma-Aldrich, Germany and were of HPLC grade.

### 4.3. Isolation of Saponins S1, S3 and S9

Three saponins were isolated from the PSM by repetitive column and flash chromatography, as reported previously [24,25]. Saponin S1 was isolated (5 mg), and its structure was confirmed by comparison of its spectral data with the literature [24]. Saponins S3 and S9 were obtained (2 mg and 4 mg, resp.) and identified by extensive spectral analyses and comparison to the literature data [25]. All reagents and solvents are obtained from Sigma-Aldrich, Germany and were of HPLC grade.

### 4.4. Analysis of DEAG and PSM

An aliquot of DEAG (100 mg) was dissolved in a 5.0 mL volumetric flask with MeOH. The qualitative analysis of flavonoids and saponins in DEAG was performed as described in [22,23]. An aliquot of PSM (10 mg) was dissolved in a 100.0 mL volumetric flask with MeOH. Quantitation of saponins in DEAG and PSM was conducted by a known validated method [23]. Both methods used ultra-high-performance liquid chromatography coupled with high-resolution electrospray ionization mass spectrometry (UHPLC–HRESIMS). The machine and the operating parameters were as reported in [22,23]. UHPLC conditions for the qualitative and the quantitative analyses were as in [22,23]. All reagents and solvents are obtained from Sigma-Aldrich, Germany and were of HPLC grade.

### 4.5. Cells

The MDBK (Madine and Darby bovine kidney) cell line (obtained from ATCC (№CCL-22, USA)) is cultured in low-glucose, 20 mM Hepes buffer (Sigma-Aldrich, Merck-Germany), Dulbecco’s Modified Eagle Medium (DMEM) (Sigma-Aldrich, Merck-Germany), with 10% growth medium and 4% maintenance medium fetal calf serum (FCS) (Sigma).

### 4.6. Cultivation of Human Coronavirus 229E in Cell Culture

This study employed the human Coronavirus 229E (HCoV-229E) (genus Alpha Coronavirus) obtained from the Bulgarian National Bank for industrial microorganisms and cell cultures. A cell culture vessel with a dense cell monolayer formed is used to prepare viral stock. Cells were inoculated with undiluted virus suspension; this was followed by incubation for 1.5 h in a thermostat (Binder) at 37 °C in order to adsorb the virus. After the adsorption of the virus, the appropriate volume of maintenance medium was added. The virus was cultivated at 37 °C. The cytopathic effect of the virus is monitored daily. On the fourth day, upon reaching 90–100% CPE of the total cell area, the vessel with the inoculum was frozen and thawed three times, and virus isolation followed. The resulting virus suspension was quantitatively characterized by the final limiting dilution method, and the infectious titer was calculated by the Reed–Muench method [26]. The viral stock was dispensed into Eppendorf dishes that were stored in cold chambers at −70 °C. 

### 4.7. Cytotoxicity Assay

A colorimetric MTT assay was applied to determine the cytotoxicity of the DEAG, the individual saponins, and the purified saponin mixture [21]. The methodology is described in a previous publication of the author’s collective [21]. A cytotoxic concentration of 50 (CC_50_) and a maximum tolerated concentration (MTC) were determined. The 50% cytotoxicity concentration (CC_50_) was calculated by regression analysis of the dose–response curves. The MTC was defined as the highest concentration at which the calculated value for cell viability equals 100%. 

### 4.8. MTT-Based Colorimetric Assay for Detection of HcoV-229E Replication Inhibition

The antiviral activity of the studied extract was determined by the MTT test developed by Mosmann [31] and modified by Sudo [32]. Two experimental designs were used in which the substances were added to the virus-infected cells at different times simultaneously with the inoculation of the virus (simultaneous treatment) and after adsorption of the virus (sequential treatment). For the experiment, we used 100 TCID_50_ (50% tissue culture infective dose) viral suspension. Our previous publication used and described the methodology when using another virus agent (SvHA) [21].

In both experimental setups, plates were processed as follows: Control cells (not infected with virus and not treated with extract)—0.2 mL of supporting nutrient medium are added to the wells designated for cell control (at least 3);Virus control (virus-infected and extract-untreated cells)—to the wells designated for virus control (at least three), 0.1 mL of supporting nutrient medium is instilled;Cells exposed to the extract—(infected with a virus and treated with different dilutions of the studied extract)—0.1 mL of the previously prepared dilutions of the extract.

The 50% effective concentration (EC_50_) was calculated by regression analysis of the dose–response curves generated from the data. The following formula calculated the selectivity index (SI): CC_50_/EC_50_.

## 5. Conclusions

From the presented results, we can assume that the active action of the extract is mainly due to its saponin content since the main secondary metabolites of the species are flavonoids and saponins. The fact that individual isolated saponins show no effect, but the purified saponin mixture showed 100% protection of infected cells proves the advantage of these saponins, in a mixture, due to the possible synergistic action of the individual components. DEAG and PSM show selective indices above 10, which, according to the literature data, is promising for their further investigation of anti-Coronavirus action.

## Figures and Tables

**Figure 1 ijms-24-16525-f001:**
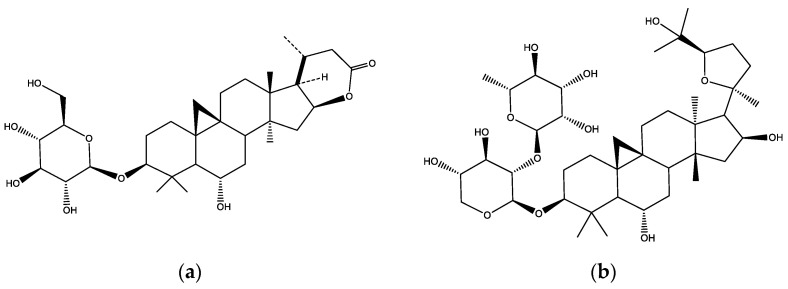
The main investigated metabolites: (**a**) structure of the analytical marker (17(*R*),20(*R*)-3*β*,6*α*,16*β*-trihydroxycycloartanyl-23-carboxylic acid 16-lactone 3-*O*-*β*-D-glucopyranoside (S1); (**b**) Astrachrysoside A (S9); and (**c**) structure of 3-*O*-[*α*-L-rhamnopyranosyl-(1→2)]-*β*-D-xylopyranosyl]-24-*O*-*α*-L-arabinopyranosyl-3*β*,6*α*,16*β*,24,25-pentahydroxy-20*R*,24*R*-cycloartane (S3).

**Figure 2 ijms-24-16525-f002:**
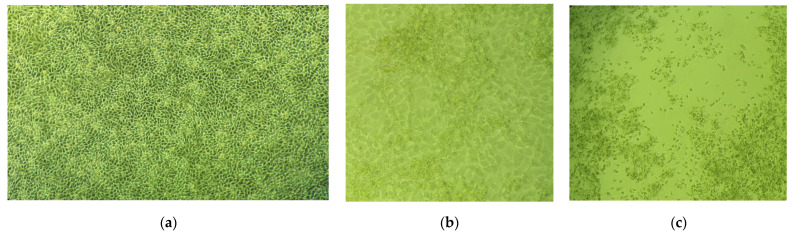
Cultivation of human Coronavirus 229E in cell culture: (**a**) cell control–uninfected cells; (**b**) manifested cytopathic effect on the second day after virus inoculation; (**c**) manifested cytopathic effect on the fourth day after virus inoculation.

**Figure 3 ijms-24-16525-f003:**
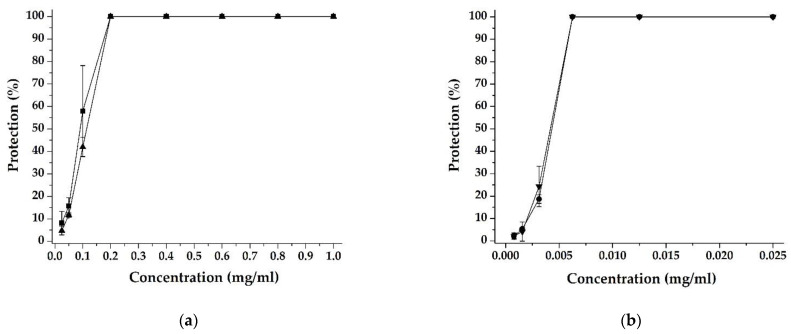
Antiviral activity according to MTT-based colorimetric assay for detection of virus replication inhibition: (**a**) DEAG added simultaneously with the inoculation of cell monolayer (
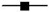
) and one and a half hours after the inoculation of cell monolayer (
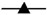
) with HCoV 229E; (**b**) PSM added simultaneously with the inoculation of cell monolayer (
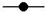
) and one and a half hours after the inoculation of cell monolayer (
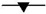
) with HCoV 229E.

**Table 1 ijms-24-16525-t001:** Data for cell viability (cytotoxicity) and cell protection (antiviral activity) by DEAG, PSM, S1, S3, and S9 were added simultaneously with, and 1.5 h after, the inoculation of the cell monolayer with HCoV-229E.

Type of Test Sample	Cell Viability (Cytotoxicity)		Cell Protection (Antiviral Activity)					
			Test Sample Added Simultaneously with Inoculation of Cell Monolayer			Test Sample Added 1.5 h after Inoculation of Cell Monolayer		
	MTC (mg/mL)	CC_50_ ^a,b^ (mg/mL)	Cell Protection (%) When the Extracts are Added in MTC ^a^	EC_50_ ^a,b^ (mg/mL)	SI ^c^	Cell Protection (%) When the Extracts Are Added in MTC ^a^	EC_50_ ^a,b^ (mg/mL)	SI ^c^
DEAG	1	1.272 (±0.11)	100	0.091 (±0.06)	13.97	100	0.114 (±0.10)	11.15
PSM	0.025	0.0748 (±0.18)	100	0.00432 (±0.14)	17.31	100	0.00418 (±0.31)	17.89
S1	0.00625	0.117 (±0.03)	n.d. ^d^	n.d. ^d^	n.d. ^d^	n.d. ^d^	n.d. ^d^	n.d. ^d^
S3	0.025	0.163 (±0.21)	n.d. ^d^	n.d. ^d^	n.d. ^d^	n.d. ^d^	n.d. ^d^	n.d. ^d^
S9	0.00625	0.104 (±0.53)	n.d. ^d^	n.d. ^d^	n.d. ^d^	n.d. ^d^	n.d. ^d^	n.d. ^d^

MTC—maximum tolerable concentration, CC_50_—cytotoxic concentration 50, EC_50_—effective concentration 50. ^a^ The results are expressed as the mean value. ^b^ The parentheses represented ±SD. ^c^ SI (selective index)—the ratio of CC_50_ and EC_50_ (SI = CC_50_/EC_50_). ^d^ n.d.—not detected.

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
