# Peer review of "Effect of a Total Extract and Saponins from Astragalus glycyphyllos L. on Human Coronavirus Replication In Vitro"

_ijms, 2023, doi:10.3390/ijms242216525_

Round 1

Reviewer 1 Report

Comments and Suggestions for Authors

Effect of total extract and saponin fractions from Astragalus glycyphyllos L. on human coronavirus replication in vitro

Authors: Аnton Hinkov * , Venelin Tsvetkov , Aleksandar Shkondrov , Ilina Krasteva , Stoyan Shishkov , Kalina Shishkova *

Abstract

Members of the family Coronaviridae cause diseases in mammals, birds, and wildlife (bats), some of which may be transmissible to humans or specific to humans. In the human population, they can cause a wide range of diseases, mainly affecting the respiratory and digestive systems. In the scientific databases, there are huge numbers of research articles about the antiviral, antifungal, antibacterial, antiviral, and anthelmintic activities of medicinal herbs and crops with different ethnobotanical background. The subject of our research is the antiviral effect of saponin fractions, purified saponin mix and methanol extract of Astragalus glycyphyllos L. In the studies conducted for the cytotoxic effect of the substances, CC50 and MТC were determined by colorimetric method (MTT assay). The virus was cultured in the MDBК cell line. As a result of the experiments carried out on the influence of substances on viral replication (using MTT-based colorimetric assay for detection of human coronavirus replication inhibition), it was found that the total extract of astragalus and the purified saponin mix inhibited 100% viral replication. The calculated selective indices are about 13 and about 18, respectively. Тhe obtained results make them promising for a preparation with anti-coronavirus action.

Comment to the authors:

This manuscript is quite interesting because it provides significant information about potential plant-based treatments for coronavirus through in vitro evaluation. However, I believe that it is still a long way from being ready for publication. There are numerous errors, and it has not been carefully prepared. Therefore, I recommend that it be rejected and resubmitted.

  1. Many related references have not been examined or cited, such as:
    • "Honeysuckle (Lonicera japonica) and huangqi (Astragalus membranaceus) suppress SARS-CoV-2 entry and COVID-19 related cytokine storm in vitro" in Frontiers in Pharmacology, 12 (2021), Article 765553.
    • "Identification of triterpenoid saponin inhibitors of interleukin (IL)-33 signaling from the roots of Astragalus membranaceus" in the Journal of Functional Foods, Volume 101, February 2023, 105418.
    • And so on.
  2. It does not contain any evidence of UHPLC-HRESIMS qualitative analysis of A. glycyphyllos.
  3. Line 86-98 lacks references to potential drug candidates against SARS-CoV-2.
  4. I do not see the statistical analysis part, and Figure 2 and Table 1 are also missing. What method was used in this study?
  5. The descriptions of the Phytochemistry of DEAG and PSM were not careful. The same applies to sections 4.2, 4.3, and 4.4. Please provide more information.
  6. There are many errors throughout the manuscript, such as in line 205 and line 216 concerning the species' names. There are too many errors, so I cannot list them all here. Please carefully review manuscript.
  7. Figure 1 is not clear in showing the configuration of the sugar. Please double-check whether the sugar is Beta or alpha."

Comments on the Quality of English Language

Extensive editing of English language required

Author Response

Point-by-point response to Comments and Suggestions for Authors by Review 1

Comment to the authors:

This manuscript is quite interesting because it provides significant information about potential plant-based treatments for coronavirus through in vitro evaluation. However, I believe that it is still a long way from being ready for publication. There are numerous errors, and it has not been carefully prepared. Therefore, I recommend that it be rejected and resubmitted.

  1. Many related references have not been examined or cited, such as:
    • "Honeysuckle (Lonicera japonica) and huangqi (Astragalus membranaceus) suppress SARS-CoV-2 entry and COVID-19 related cytokine storm in vitro" in Frontiers in Pharmacology, 12 (2021), Article 765553.

Answer: The representatives of the Fabaceae family are a huge number. It is possible that the agent you mentioned blocks the entry of SARS-CoV-2. In our study, we used an extract and saponin mix from the agent Astragalus glycyphyllos, which has not been tested for antiviral activity. In the introduction, we have described some actions of representatives of the family, but for this representative - Astragalus glyciphyllos, information is scarce. In addition, experiments were conducted with HCoV – 229Е, belonging to the genus alphacoronavirus.

  • "Identification of triterpenoid saponin inhibitors of interleukin (IL)-33 signaling from the roots of Astragalus membranaceus" in the Journal of Functional Foods, Volume 101, February 2023, 105418.

Answer: Regarding the immunomodulating activity of another member of the family - Astragalus membranaceus - our team has no claims. Our research is on the antiviral activity of Astragalus glycyphyllos and I think it is on the topic set by the journal.

  • And so on.

  1. It does not contain any evidence of UHPLC-HRESIMS qualitative analysis of A. glycyphyllos.

Answer: Supplementary figures are added. The analytical methods used have already been validated and published. The information on its qualitative analysis coincides with literature. Relevant citations are given, to eliminate repetition of previously published materials. The focus of this work is not analysis, but evaluation of possible antiviral activity.

  1. Line 86-98 lacks references to potential drug candidates against SARS-CoV-2.

Answer: I think potential candidates are cited in the literature as number 14.

  1. I do not see the statistical analysis part, and Figure 2 and Table 1 are also missing. What method was used in this study?

Answer: I'm afraid I don't understand what exactly is missing. Figure 2 is on page 5 and Table 1 is on page 7. The methods used are described on page 9 with relevant citations. Standard deviations are described in Table 1.

  1. The descriptions of the Phytochemistry of DEAG and PSM were not careful. The same applies to sections 4.2, 4.3, and 4.4. Please provide more information.

Answer: The descriptions were re-checked, and corrected, accordingly. Relevant citations are given, to eliminate repetition of previously described and validated methods.  

  1. There are many errors throughout the manuscript, such as in line 205 and line 216 concerning the species' names. There are too many errors, so I cannot list them all here. Please carefully review manuscript.

Answer: On line 205 I think the species is spelled correctly, and on line 216 the information is accompanied by the relevant bibliographic reference.

  1. Figure 1 is not clear in showing the configuration of the sugar. Please double-check whether the sugar is Beta or alpha."

Answer: The figures were corrected. The compounds are following the original publications: https://www.mdpi.com/2218-1989/13/7/857; https://www.tandfonline.com/doi/abs/10.1080/14786419.2018.1491040

Reviewer 2 Report

Comments and Suggestions for Authors

The manuscript by Hinkov et al. is a interesting scientific article. The topic of natural, and therefore potentially safer than synthetic, antiviral compounds is important. The results obtained are worth to present, however the article needs improvements. Some suggestions and issues to explain are listed below:

1) What was the purity of saponins S1, S3 and S9?

2) Plant-derived compounds often have antioxidant properties, which may affect measurements using MTT. Was there any interaction found for the samples tested isolated from the plant? Was this taken into account when calculating cell survival?

3) The "Conclusions" section should be slightly expanded with a statement of what fraction of saponins was most beneficial and at what concentration and to what extent it protected the cells.

4) The description of the images in Figure 3 is enigmatic. What metabolites are being referred to? What were the cells treated with? What is the scale of the photos? Shouldn't photos for all extracts (DEAG, PSM, S1, S3 and S9) be included?

5) The number of results presented is relatively small, to enrich the study, I would include, for example, data from chromatographic analyses of DEAG, PSM, S1, S3 and S9. 

6) Could the authors propose a protective mechanism for the extracts? Do they act on the virus, or do they enhance the cells in some way (e.g., making it more difficult for the virus to enter)?

7) Has the toxicity of the extracts themselves on cells been studied? Wouldn't they be toxic to the cells themselves at higher concentrations?

Author Response

Point-by-point response to Comments and Suggestions for Authors by Review 2

Comments and Suggestions for Authors

The manuscript by Hinkov et al. is a interesting scientific article. The topic of natural, and therefore potentially safer than synthetic, antiviral compounds is important. The results obtained are worth to present, however the article needs improvements. Some suggestions and issues to explain are listed below:

  • What was the purity of saponins S1, S3 and S9?

Answer: An explanation on the purity of the saponins was added in the text.

  • Plant-derived compounds often have antioxidant properties, which may affect measurements using MTT. Was there any interaction found for the samples tested isolated from the plant? Was this taken into account when calculating cell survival?

Answer: I agree that it is possible for extracts to affect oxidative stress, as indeed many plant extracts do. However, our goal is another, namely an anti-virus action. It turned out that the extracts in a non-toxic concentration affected 100% the replication of this virus, in contrast to the results obtained in the experiments with HSV. This is rare and we will continue research with in vivo trials.

  • The "Conclusions" section should be slightly expanded with a statement of what fraction of saponins was most beneficial and at what concentration and to what extent it protected the cells.

Answer:  MТC for the individual saponins as well as the saponin mixture are presented in Table 1. Since they are not active in terms of viral replication, we have also drawn the relevant conclusions. The text was revised, and accordingly corrected to clear the conclusion.

  • The description of the images in Figure 3 is enigmatic. What metabolites are being referred to? What were the cells treated with? What is the scale of the photos? Shouldn't photos for all extracts (DEAG, PSM, S1, S3 and S9) be included?

Answer: Pictures refer to virus cultivation section. We find it interesting that a coronavirus was successfully cultivated in the MDBК cell line. Cells were not treated with any metabolite. We show a cytopathic effect at the corresponding hour. We have successfully cultured the infectious virus and have a distinct cytopathic effect. When treating infected cells with DEAG, the picture will look like control cells. The same goes for the saponin mix. The percent protection of the extract and the saponin mixture is 100When treated with the individual saponin fractions, the picture is similar to figure 2 (c). If adding these photos will not complicate the article, we will add them.

  • The number of results presented is relatively small, to enrich the study, I would include, for example, data from chromatographic analyses of DEAG, PSM, S1, S3 and S9. 

Answer: Chromatographic studies are correctly indicated and cited in the bibliographic reference. Since the methods for used phytochemical analysis have already been reported, the analysis has only a supportive role. Therefore, chromatograms are given as Supplementary figures because the focus of the work is an antiviral study.

  • Could the authors propose a protective mechanism for the extracts? Do they act on the virus, or do they enhance the cells in some way (e.g., making it more difficult for the virus to enter)?

Answer: Studies to prevent adsorption and penetration of the virus have not been done. The fact that they affect its replication is encouraging enough. When the infected cells were treated in parallel and sequentially, the results were almost the same, leading to the conclusion that the extracts were unrelated to the extracellular virion.

  • Has the toxicity of the extracts themselves on cells been studied? Wouldn't they be toxic to the cells themselves at higher concentrations?

Answer: Of course they are researched. Based on these studies, MТC and CC50 were determined. Any antiviral study begins with cytotoxicity experiments. The results are presented in Table 1.

Round 2

Reviewer 1 Report

Comments and Suggestions for Authors

I am satisfied with this revision. I recommend accepting it.

Comments on the Quality of English Language

Minor editing of English language required

Author Response

Thank you for your time and support!

Reviewer 2 Report

Comments and Suggestions for Authors

Thank you for your answers. I think the manuscript is ready to be published. 

Author Response

Thank you for your time and support!